

# The role of biotic factors during plant establishment in novel communities assessed with an agent-based simulation model

Janina Radny and Katrin M. Meyer

Department of Ecosystem Modelling, Georg-August-Universität Göttingen, Göttingen, Germany

## ABSTRACT

**Background**. Establishment success of non-native species is not only influenced by environmental conditions, but also by interactions with local competitors and enemies. The magnitude of these biotic interactions is mediated by species traits that reflect competitive strength or defence mechanisms. Our aim was to investigate the importance of species traits for successful establishment of non-native species in a native community exhibiting biotic resistance in the form of competition and herbivory.

**Methods**. We developed a trait-based, individual-based simulation model tracking the survival of non-native plants in a native community. In the model, non-native plants are characterized by high or low values of competition and defence traits. Model scenarios included variation of initial number of non-natives, intensity of competitive interaction, density of herbivores and density as well as mixture of the native community.

**Results**. Traits related to competition had a much greater impact on survival of non-native species than traits related to defence. Survival rates of strong competitors never fell below 50% while survival of weak competitors averaged at about 10%. Weak competitors were also much more susceptible to competitive pressures such as community density, composition and competition intensity. Strong competitors responded negatively to changes in competition intensity, but hardly to composition or density of the native community. High initial numbers of non-native individuals decreased survival rate of strong competitors, but increased the survival rate of weak competitors. Survival under herbivore attack was only slightly higher for plants with high defensive ability than for those with low defensive ability. Surprisingly, though, herbivory increased survival of species classified as weak competitors.

**Discussion**. High survival rates of strong non-native competitors relate to a higher probability of successful establishment than for weak competitors. However, the reduced survival of strong competitors at high initial numbers indicates a self-thinning effect, probably mediated by a strongly competitive milieu. For weak competitors, our model emphasizes positive effects of high propagule pressure known from field studies. General effects of herbivory or defence abilities on survival were not supported by our model. However, the positive effect of herbivory on survival of weak competitors indicated side effects of herbivory, such as weakening resident competitors. This might play an important role for establishment of non-natives in a new community.

Corresponding author
Katrin M. Meyer,
Katrin.Meyer@forst.uni-goettingen.de,
kmeyer5@uni-goettingen.de

## INTRODUCTION

In response to current climate change, range borders and distribution patterns of many species have shifted along with changes in environmental conditions (*Chen et al., 2011*; *Maggini et al., 2011*; *Parmesan & Yohe, 2003*; *Walther et al., 2002*). The abiotic and biotic environment in novel habitats plays an important role for capturing and eventually predicting range shift dynamics of species (*Berg et al., 2010*). On a large scale range shifts are correlated with bioclimatic and environmental factors and whether a plant species is able to reach a novel habitat. Thus, range expansion of a species is a continuous process, acting on a large-scale landscape. However, realization of range expansion is determined by a series of successful local establishment events of individuals (*Bakkenes et al., 2002*), that build up viable populations in the novel habitat (Fig. 1). Local establishment success strongly depends on biotic interactions with the resident community. Such interactions include for example competition with resident plants or herbivore attack (*Levine, Adler & Yelenik, 2004*). In fact, diverging trends in speed, extent and directions of species range shifts may be not only an expression of their ability to reach a novel habitat, but also to persist in novel communities (*Lenoir et al., 2010*; *Maggini et al., 2011*). These local negative processes can add up and might eventually prevent further range shifts. Thus, although a species range is typically described on large, often continental scales, it is limited by small-scale processes at the borders of the range. This is in line with the claim to consider dispersal and local establishment success of species and individuals in order to gain insights on range dynamics (*Guisan & Thuiller, 2005*; *Pearson & Dawson, 2003*; *Wisz et al., 2013*). In a novel habitat, the non-native species will face novel interactions with the resident community. The magnitude of the effect of these interactions is partially determined by the ability of the novel species to respond to the biotic pressures (*Kempel et al., 2013*; *Roux et al., 2012*; *Wisz et al., 2013*). This ability is mediated by plant traits. Despite the general acknowledgement of the importance of traits and biotic pressures for establishment success, the interaction of these factors is not well studied in the context of range expansion.

We constructed an individual-based simulation model to scrutinize how these factors together affect success or failure of plant establishment in a novel community. Field surveys are generally incapable of capturing failed invasion attempts (*Zenni & Nunez, 2013*), but modelling allows for comprehensive systematic modifications and full control of the setup. Data from a preceding greenhouse experiment (*Radny et al., in press*) were used as a basis for model construction and parametrization (see 'Parametrization').

Early establishment of populations in novel habitats in a local context is an essential first step in range shift dynamics of a species. Therefore, in this model we focus on such biotic interactions that operate on local scales and with immediate contact between the interacting entities. These biotic interactions are enemy attack and competition for resources between neighbouring plants.

Biotic pressures such as competition and enemy attack may hamper establishment success of a novel species in a community by decreasing individual fitness. The causes are obvious: when mutually contested resources are captured by competing species, the inferior competitor can use less of the contested resource for its own growth and reproduction.

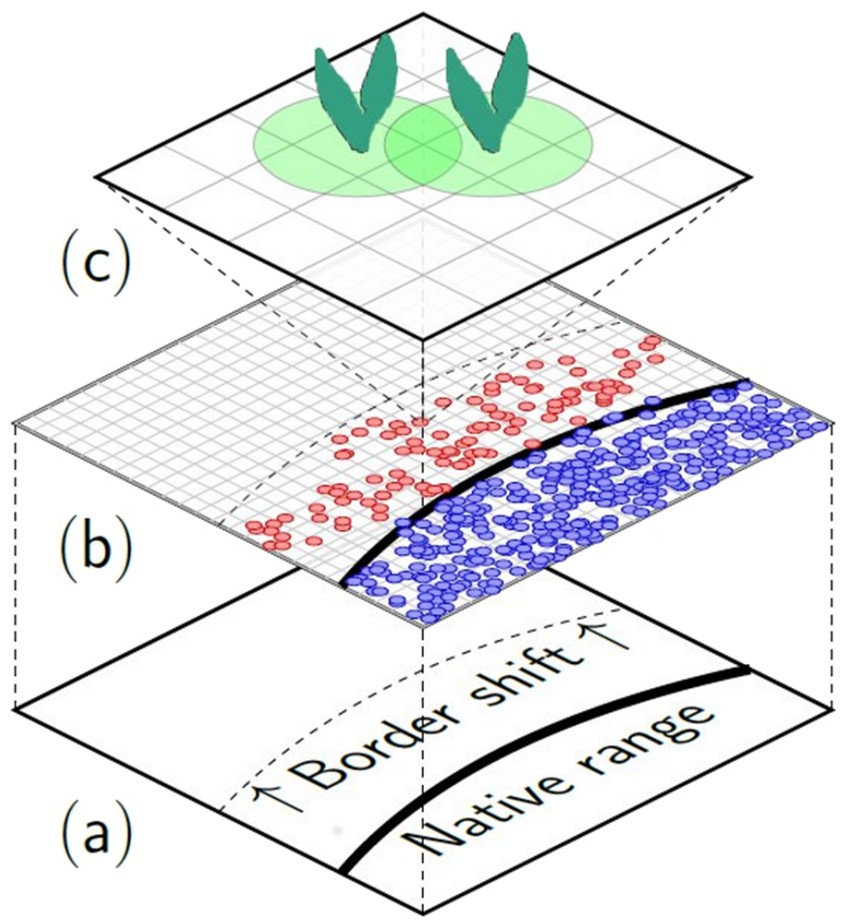

**Figure 1** **Overview of the scale-dependent relationship between range shift and local establishment of plants.** Range shift occurs on a large scale and at species level (A). This movement can be disaggregated into successful establishment events of local satellite populations beyond the native range border (red dots, B). Local persistence depends on the fitness of individuals in this local population. This is influenced by direct interactions, e.g., with neighbouring competitors (plant symbols) or resident herbivores (not shown), on this local scale (C).

Eventually, it may die when resource uptake falls below a metabolic threshold (*Lin et al., 2012*; *Schmitz, 2000*). Enemy attack can strongly weaken a plant through damage of plant tissue or organs and even result in lethal consumption. Loss of tissue may pose an additive negative effect on plants under competition (*Heard & Sax, 2013*; *Kim, Underwood & Inouye, 2013*; *Kuijper, Nijhoff & Bakker, 2004*). In fact, invasion biology studies have often pointed out that very effectively expanding, i.e., invasive, non-native species suffer less from enemy attack than native or non-invasive non-native species (*Cappuccino & Carpenter, 2005*; *Engelkes et al., 2008*; *Matter et al., 2012*; but see *Heard & Sax, 2013*). This effect, referred to as enemy release, is one line of argument to explain invasiveness of non-native plant species in intercontinental invasions (*Joshi & Vrieling, 2005*; *Keane & Crawley, 2002*) since lately enemy release is also studied in intracontinental range shifts,

motivated by long-distance dispersal of plant species (*Engelkes et al., 2008*; *Harvey et al., 2010*; *Nehrbass et al., 2007*).

Traits of species mediate interactions and are often used to describe community composition in classical community ecology (*McGill et al., 2006*). Intercontinental invasion biology relies on certain traits that are shared by successful invaders to inform risk assessments. However, the trait set of the "perfect invader" remains to be found (*Speek et al., 2011*). Further, it is largely unclear how strongly trait sets vary under different interaction regimes. For our model, we use the framework of biotic resistance composed of resource competition and herbivore attack (*Levine, Adler & Yelenik, 2004*). We describe novel species by a set of traits to respond to these components of biotic resistance. Our central model aim is to assess how non-native species with different trait profiles can establish in a novel community despite negative biotic interactions.

The traits used in the model reflect the ability of a plant to (a) compete with neighbours and (b) defend against enemies. Based on size-symmetric competition models (*Connolly & Wayne, 1996*; *Damgaard & Weiner, 2008*), we use seed mass as a measure of plant size and as the trait representing competitive ability (*Metz, Nisbet & Geritz, 1992*). Other traits have also been associated with competitive strength (*Goldberg, 1996*), but plant size is a straightforward measure (*Domingos, 1999*), which is well supported (*Aikio, 2004*; *Weiner & Damgaard, 2006*). Mechanical barriers and toxins belong to the traits a plant can use to prevent herbivore attack, and often plants build up a complex defence syndrome of multiple traits (*Agrawal & Fishbein, 2006*). For our purpose, the form of defence mechanism is not of interest. Thus, we implemented the defence mechanism as the stochastic ability of enemy repulsion.

In summary, the aim of the model presented here is to systematically investigate how local establishment success of non-native species is influenced by biotic resistance of the native community, by competition and defence traits of non-natives, and by the interaction of biotic resistance and traits. We present a comprehensive analysis of the consequences of a broad range of parameters representing biotic resistance and species traits for non-native species survival and thus local establishment.

## MATERIALS & METHODS

### Model description

Our model description follows the ODD protocol (*Grimm et al., 2010*; *Grimm et al., 2006*). Model structure is partly based on (*Lin et al., 2012*).

### *Purpose*

The model compares first generation establishment success of generic range expanding species in a novel community, based on a hypothetical ecological profile. The ecological profile is expressed by combinations of high or low ability to compete with neighbouring plants and defend against insect herbivores. We assign seed size to competitive ability and repulsion of herbivores to defence ability (Table 1). In the following, generic species are called "species" for better readability.

**Table 1  Parameter values for traits of non-native plants.** Seed weight was directly measured and averaged from plant material used in the preceding experiment (*Radny et al., in press*) and based on the species *S. capensis* and *B. fasciculatus*. We chose maximum biomass relative to the native community. Repulsion of herbivores was averaged over all strongly and weakly defended species in the preceding experiment respectively.

| Parameter | Unit | Value | | Source |
|---|---|---|---|---|
| *Competitive strength values* | | *Strong competitors* | *Weak competitors* | |
| Seed weight | mg | 3 | 1 | *Radny et al. (in press)* |
| Maximum biomass | mg | 15,000 | 4,000 | |
| *Defensive strength values* | | *Strong defenders* | *Weak defenders* | |
| Repulsion of herbivores | % | 70 | 30 | *Radny et al. (in press)* |

**Table 2  Parameter values for traits of native plants.** Note that maximum age is aligned for all species. Reproduction and dispersal values are not considered for native species as we were not interested in the fate of native species in this approach. Parameter values were either directly retrieved or approximated (∗) from the available literature. The composition of the native community in uneven mixtures is based on seed bank values. Initial seed density for *F. rubra* (∗∗) has been approximated from vegetation cover (30%) as retrieved from *Edwards & Crawley (1999)*, assuming that all seeds germinate and build the entire native community.

| Species | Parameter | Unit | Value | Source |
|---|---|---|---|---|
| *Plantago lanceolata* | Seed weight | mg | 1.4 | *Schmitt, Niles & Wulff (1992)*∗ |
| | Maximum biomass | mg | 8,000 | *Janeček, Patáčová & Klimešová (2014)* |
| | Repulsion of herbivores | [] | 0.4 | *Radny et al. (in press)* |
| | Initial seeds | $\frac{1}{m^2}$ | 85 | *Edwards & Crawley (1999)* |
| *Hypericum perforatum* | Seed weight | mg | 0.09 | *Fox et al. (1999)* |
| | Maximum biomass | mg | 5,800 | *Vilà, Gómez & Maron (2003)* |
| | Repulsion of herbivores | [] | 0.9 | *Radny et al. (in press)* |
| | Initial seeds | $\frac{1}{m^2}$ | 98 | *Matus, Papp & Tóthmérész (2005)* |
| *Agrostis capillaris* | Seed weight | mg | 0.06 | *Stukonis & Slepetys (2013)* |
| | Maximum biomass | mg | 4,000 | *Ehlers (2011)* |
| | Repulsion of herbivores | [] | 0.3 | *Radny et al. (in press)* |
| | Initial seeds | $\frac{1}{m^2}$ | 1,343 | *Edwards & Crawley (1999)*∗ |
| *Festuca rubra* | Seed weight | mg | 0.77 | *Larson, Anderson & Newton (2001)* |
| | Maximum biomass | mg | 12,000 | *Corbin & D'Antonio (2004)*∗ |
| | Repulsion of herbivores | [] | 0.3 | *Radny et al. (in press)* |
| | Initial seeds | $\frac{1}{m^2}$ | 654 | *Edwards & Crawley (1999)*∗∗ |

### Entities, state variables, and scales

We implemented two general types of agents, plants and herbivores. Plants are non-mobile agents, characterized by their x-y-position and values related to growth, reproduction and defence (Tables 1 and 2). On initialization, the plants are represented as seeds and hatch in the first step. We distinguish two general types of plants, native and non-native, and twelve species, of which four are native and eight are non-native. Parameters have been retrieved from a preceding experiment (*Radny et al., in press*) and from the literature (see 'Model parametrization and validation').

The simulated world comprises of a homogenous $100 \times 100$ cell grid. Each grid cell represents 1 cm$^2$, thus we model a $1 \times 1$ m plot. To avoid edge effects, the world is a torus, i.e., opposing edges are connected (*Grimm & Railsback, 2005*).

Each time step is representing one day. A simulation comprises 76 days, i.e., the average lifespan of a non-native species in the preceding experiment. The simulation was aborted earlier if no non-native individuals were left on the grid.

### Process overview and scheduling

During each time step, the simulation routine is exerted in the following order: herbivores appear and feed on random plants, competition intensity is calculated for each plant individual, plants grow according to this competition intensity and plants age. After 76 days, the plants produce and disperse seeds and die. Dead plants are removed from the grid. The process schedule is visualized in Fig. 2.

### Basic principles

We follow the assumption that community composition can at least partly be related to biotic interactions and that biotic interactions are closely connected to functional traits of species (*McGill et al., 2006*). We apply a basic profile of functional traits related to the ability of species to (a) compete with neighbours and (b) defend against enemies. We also address biotic resistance, i.e., negative impacts of resident herbivores and competitors on non-native plants, which can influence local establishment success (*Kempel et al., 2013*).

### Emergence

Several patterns related to individual plant development and population dynamics emerge from the model. Examples include spatial patterns of plant individuals, population dynamics of non-native and native plants, reproductive output, and plant size distributions. For the purpose of this paper, we focus on initial and final number of individuals of non-native species per time step as a basis for deriving survival rate in the establishment phase (see 'Scenario analysis').

### Interactions

We explicitly model competitive interactions between neighbouring plants via the Zone-Of-Influence (ZOI) approach (*Lin et al., 2012*; *Weiner & Damgaard, 2006*). Furthermore, herbivores interact with plants by consuming parts of the aboveground biomass of the encountered plants.

### Stochasticity

Plant individuals are initialized with random $x$- $y$-coordinates. To reflect individual deviance from population mean values, initial biomass of plants is calculated as seed mass multiplied by a random number, drawn from a normal distribution with a mean of 1 and a standard deviation of 0.1. Instead of defining fixed relative growth rates (rgr), plants determine their own growth rate through

$$ln(maximum\,biomass) - ln(initial\,biomass)/maximum\,age \qquad (1)$$

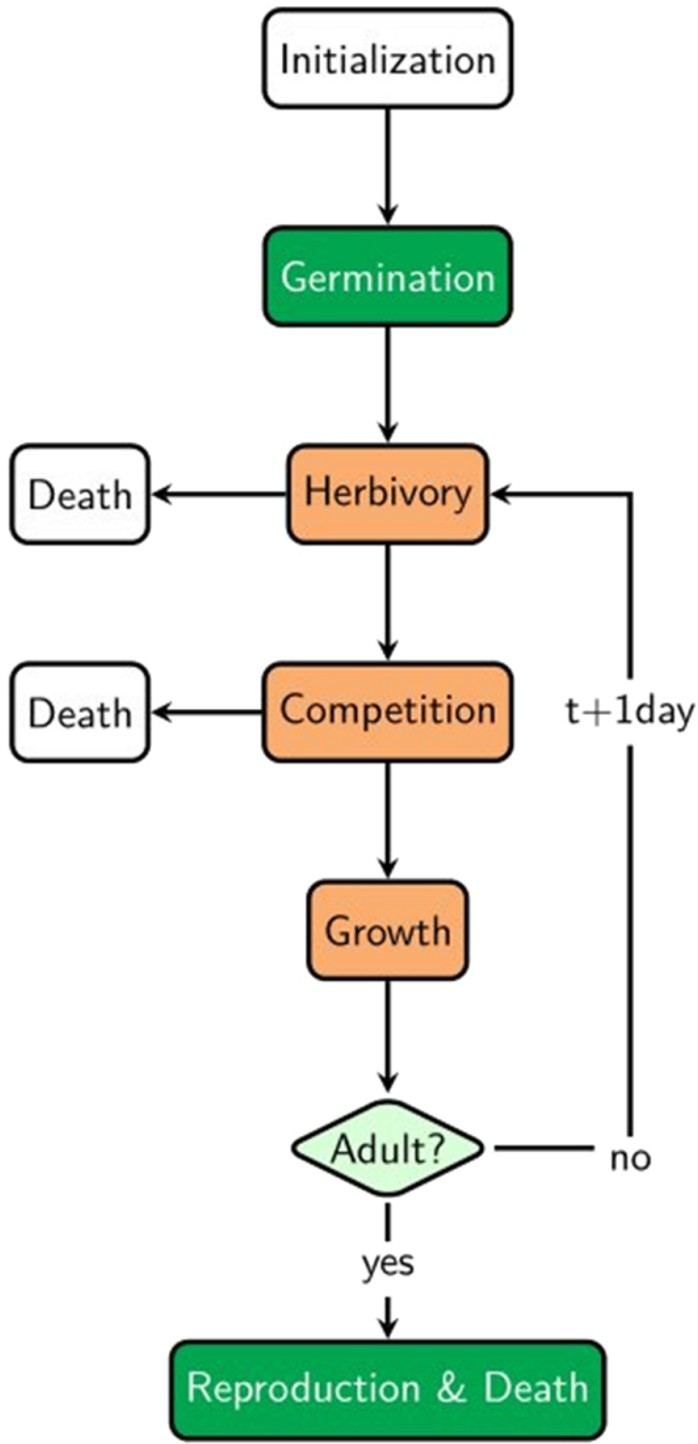

**Figure 2 Order of executed routines during each simulation.** Green-coloured routines are executed only once; orange-coloured routines are executed on a "daily" basis, i.e., at each time step. Herbivory and Competition can lead to individual death of plants.

**Table 3 Overview of components of model scenarios.** Model scenarios were put together by choosing one level per component. All possible combinations of levels were run in our model analysis.

| Component of model scenario | Number of levels | Levels |
| --- | --- | --- |
| Trait profiles | 4 | high competitive and high defensive (HiAll),high competitive and low defensive (HiComp), low competitive and high defensive (HiDef), low competitive and low defensive (LowAll) |
| Initial population size of non-natives | 6 | 8, 16, 32, 64, 128, 256 |
| Competition intensity or asymmetry (theta) | 3 | 0, 0.5, 1 |
| Community density | 2 | 300 plants (low), 1,100 plants (high) |
| Mixture of the native community | 2 | same initial density of all species (even), species initial densities differ based on reported seed bank sizes and germination rates (literature based) |
| Herbivore density | 3 | no herbivores (control), 3.2 herbivores per $m^2$ (low), 18.75 herbivores per $m^2$ (high) |

(*Hunt, 1982*). This approach propagates the stochasticity involved in initial biomasses to the distribution of relative growth rates. Daily density of herbivores is stochastic, as grasshopper density is multiplied with a stochastic number of meals per herbivore (see Submodels: Herbivory). Success of each foraging attempt of herbivores on a given plant is stochastic, based on the repulsion value of the plant.

### Initialization

Each simulation run starts with placement of seeds on the grid at random xy-positions. Initial native community population size is based on plant densities in the preceding greenhouse experiment (*Radny et al., in press*). In that experiment, planting pots of $18 \times 18$ cm carried either 12 or 44 native individuals, i.e., $\sim$300 or 1,100 individuals per $m^2$. Mixtures of native species are based on either the experiment or approximated literature values (Table 2) depending on the simulation scenario (Table 3). For experiment-based mixtures, each native species is represented with the same number of individuals, called "even mixture" hereafter. For literature values, we use the reported seed bank size of the respective species as initial numbers (Table 2), called "seedbank mixture" hereafter. Per simulation scenario, there is one non-native species mixed into the native community. Different simulation scenarios were run with initially 8, 16, 32, 64, 128 and 256 seeds of non-native species. These scenarios reflect different levels of introduction efforts or proximity to core range with higher population densities (*Kolar & Lodge, 2001*).

### Input data

The model does not require any input data beyond the model parameters.

### Submodels

*Herbivory.* Herbivore agents are created at the beginning of each time step (Fig. 2). The grasshopper *Locusta migratoria* is used as herbivore model species due to its wide distribution (*CABI, 2013*) and generalist feeding behaviour (*Macel et al., 2005*; *Schmitz & Booth, 1997*). We had also chosen this species as herbivore in the preceding experiment (*Radny et al., in press*). Herbivore density is varied based on literature values and density
in the experiment, ranging from 3.2 grasshoppers per m² (literature value *Ledergerber, Thommen & Baur, 1997*) to 75 grasshoppers per m² (experimental value, based on *Morriën, Engelkes & Van der Putten, 2011*). One scenario is implemented without herbivores (Table 3). Here, neither native nor non-native species experience herbivory.

*L. migratoria* consumes 50–70 mg of plant material per day (*Simpson & Abisgold, 1985*). Consumption does not happen all at once, but in 7 to 10 meals of about 7.5 mg each. In a natural setting, it is not likely that a highly mobile insect is feeding on the same plant individual all day. Thus, we implemented in-between meal movement. To reduce computing effort, we created one dummy herbivore individual per meal. The number of dummy herbivore individuals that represent one herbivore was determined by multiplying each herbivore by a random number drawn from the interval [7, 10]. This may, for instance, lead to eight dummy herbivore individuals representing eight meals of herbivore A plus ten dummy herbivore individuals representing ten meals of herbivore B and so forth. The movement of the dummy herbivore individuals reflects in-between meal movement of the original herbivores. This simplification is justified, because we are not interested in the fate of individual herbivores, but just their effect on plant biomass. This multiplication allows us to model a process which is operating at a finer temporal scale than the global time step (one day).

Each dummy herbivore is initialized at random x-y-coordinates and approaches a random plant in a radius of 50 patches. To consume the approached plant, a random number drawn from the interval [0, 1] must be greater than the repulsion value of the respective plant. A high repulsion value thus represents high defence of a plant, e.g., through physical barriers or toxins. If dummy herbivores are not successful with foraging, they leave that meal out and are removed from the grid. This procedure still adequately represents the overall effect of herbivory on plants. All successful dummy herbivores are removed directly after foraging. Thereby, herbivore density fluctuates daily. As we observed in our greenhouse experiment (*Radny et al., in press*), a plant dies in the model if more than 90% of its current biomass is consumed by herbivores, and it is removed from the grid.

*Plant competition and growth.* Plants interact competitively with their neighbours. To model competition, we follow the Zone-Of-Influence approach (*Berger et al., 2008*; *Lin et al., 2012*; *Weiner et al., 2001*; *Weiner & Damgaard, 2006*). The ZOI of a plant is represented by a circular area $A_{ZOI}$ with radius $r$. Based on *Lin et al. (2012)*, radius $r_i$ is allometrically related to biomass $B$ of plant $i$ at time $t$:

$$r_i = B_i(t)^{3/8} * \sqrt{\frac{1}{\pi}}. \tag{2}$$

Note that *B(t)* is determined after herbivore attack. Based on *Lin et al. (2012)*, plant growth in the next time step is determined by the relative plant growth rate *rgr*, the area of the ZOI $A_{ZOI}$, current biomass $B$ and maximum biomass $B_{max}$:

$$\frac{dB}{dt} = rgr * A_{Zoi} * \left(1 - \left(\frac{B}{B_{max}}\right)^{1/4}\right). \tag{3}$$

In most cases, the ZOI of a plant covers several grid cells. Thus, $A_{ZOI}$ consists of the sum of the area of the grid cells $c$ that are occupied. Equation (3) applies when a plant is growing without neighbours, i.e., none of the cells within $A_{ZOI}$ is occupied by another but the focal plant. When two or more plants are neighbours, i.e., their ZOIs overlap in at least one cell, plant $i$ calculates its effective area $A_{eff,i}$ as the sum of the area of the grid cells $c$ it occupies, each weighted by the proportion of its current biomass $B_i(t)$ relative to the total biomass of all overlapping plant individuals $j$ to $k$ in this grid cell, modified by the degree of asymmetry of competition $\Theta$ (*Lin et al., 2012*; *Weiner & Damgaard, 2006*):

$$A_{eff,i}(t) = \sum_{c_i} \frac{B_i(t)^{\Theta}}{\sum_j^k B_j(t)^{\Theta}} dc \qquad (4)$$

where $dc$ is the area of the respective grid cell c. In our case, all grid cells have the same area. Thus, instead of using $dc$ as in Eq. (4), the summed area of the grid cells of the ZOI of plant $i$ can be rearranged and expressed in terms of the biomass of plant $i$ (*Lin et al., 2012*):

$$A_{eff,i}(t) = B_i(t)^{3/4} \sum_{c_i} \frac{B_i(t)^{\Theta}}{\sum_j^k B_j(t)^{\Theta}}. \qquad (5)$$

We use the degree of asymmetry as a measure of competition intensity. Intensity of competition determines how the contested resources at any patch are shared among the competing plants, depending on their biomass relative to the other competitors. With $\Theta = 0$, resources are shared equally among plants regardless of their biomass. Increasing $\Theta$ leads to an increasing weight of biomass for capturing contested resources in a shared cell. With $\Theta = 1$, a larger plant receives a larger share of the contested resources than a smaller plant, proportional to its biomass (perfect size-symmetry).

Mathematically, $\Theta$ can be located between 0 and $\infty$ (*Schwinning & Weiner, 1998*), but for our purpose, we use $\Theta = 0$, 0.5 and 1 in different model runs.

Plant growth under competition is implemented following *Lin et al. (2012)* as:

$$\frac{dB}{dt} = rgr * A_{eff} * \left(1 - \left(\frac{B}{B_{max}}\right)^{1/4}\right). \qquad (6)$$

The difference between Eqs. (3) and (6) is that either all resources within the zones of influence are considered to calculate growth regardless of neighbouring competitors ($A_{zoi}$), or that only the share of resources captured after competition with neighbours are considered to calculate growth ($A_{eff}$).

If growth and resource uptake fall below the threshold of 0.05 of $B^{3/4}$, the plant cannot serve its metabolic costs and dies (*Lin et al., 2012*; *Schmitz, 2000*). Overall, plant growth is strongly determined by plant biomass in our model (Fig. S1).

*Plant reproduction and dispersal.* When plants reach maturity, i.e., maximum age, they produce seeds as a function of their final biomass. So under strong competition, final biomass does not necessarily match maximum biomass. In our greenhouse experiment (*Radny et al., in press*), we obtained a positive correlation between aboveground biomass

and seed mass, i.e., total mass of all seeds produced per plant. We use this empirical correlation to determine seed mass per plant in the model, i.e., a ratio of 0.41 mg seed mass per mg shoot biomass for strong competitors and 0.65 mg seed mass per mg shoot biomass for weak competitors. The number of seeds per plant is implemented in the model as seed mass produced by the respective plant divided by average weight of a single seed (Table 2). We included stochastic variation of number of seeds of ±10%. To reduce computational ballast, the number of seeds produced is reduced by winter mortality, including seed predation, and germination probability is applied. Thus, only seeds that survive winter and germinate in the next year are explicitly created. Seeds are then dispersed in a random angle with dispersal distance following a Weibull distribution to allow for long-distance dispersal (*Colbach & Sache, 2001*; *Paradis, Baillie & Sutherland, 2002*). Number of seeds remaining on or being dispersed beyond the parental patch is recorded together with dispersal distance of each seed for future scaling-up of the model.

## Model parametrization and validation

Model parameters were partly derived from literature and partly from a preceding greenhouse experiment (*Radny et al., in press*). The greenhouse experiment was conducted at the NIOO-KNAW in Wageningen, The Netherlands. There, we planted microcosms of 18 ×18 cm with a native community of four plant species mixed with one non-native species per microcosm. The non-native species in the experiment differed in life-history traits related to competitive and defensive strength. In a fully crossed design, non-native species were exposed to two densities of natives to create different intensities of competitive pressures. Low density microcosms were planted with 12 individuals of native species and four individuals of one non-native species. High density microcosms were planted with 44 individuals of the native species and four individuals of one non-native species. Half of the microcosms were exposed to herbivory by two generalist herbivore species, *Locusta migratoria* (L.) and *Mamestra brassicae* (L.). We harvested the microcosms when non-native individuals had finished their seed production and recorded dry aboveground biomass, seed mass, and other performance parameters (*Radny et al., in press*).

For parameterization of non-native species in the model, we used data on the Mediterranean grass species *Stipa capensis (Thunb.)* and *Bromus fasciculatus (C.Presl)* from our experiment and from the literature (Table 1). The model differentiates between four trait profiles of generic non-native species: strong versus weak competitors combined with strong versus weak defenders.

Competitive strength is represented by average seed weight of *S. capensis* for strong competitors and *B. fasciculatus* for weak competitors (Table 1). Maximum biomass was assigned as a value relative to the native community (Table 2). Additionally, we derived the fitness measure biomass-seed mass ratio from the preceding experiment by dividing total seed mass by total aboveground biomass of a species. This ratio was different for strong and weak competitors in the experiment and thus serves as another distinctive index for competitive strength (Table 1).

Defensive strength was modelled as repulsion of herbivores reflecting the chance of a plant to avoid herbivore attack. Repulsion values were derived from the percentage

of individuals without or less (>5%) clearly visible leaf damage of all individuals of a species that were exposed to herbivory during the preceding experiment. Values for strong defenders were obtained from *B. fasciculatus*, whereas values for weak defenders were obtained from *S. capensis* (Table 1).

The parameterization of the native community in the model is based on the species in the experimental native community, i.e., *Agrostis capillaris* (L.), *Festuca rubra* (L.), *Hypericum perforatum* (L.) and *Plantago lanceolata* (L.). These four species are fairly widespread over Europe (*Roscher et al., 2004*), and thus range-expanding plants from southern latitudes are likely to encounter these species as interaction partners. With the exception of repulsion of herbivores which was based on the overall observations of high and low attacks in the preceding experiment, parameters used in the model for native species were based on literature data (Table 2).

In terms of validation of our model, no independent data were available for direct comparison with model outputs. Therefore, we discussed comparisons of model outputs with the results of our preceding experiment and conducted a scenario analysis (Table 3) and a sensitivity analysis to systematically explore the parameter space to explain as much output variation as possible.

## Scenario analysis

The major goal of our simulation model was to investigate the local establishment of non-native species in a novel community, based on different trait configurations. To assess local establishment, we determined final population size, i.e., the number of reproducing adults at the end of a simulation run. To account for scenario-based differences in initial population sizes, we calculated survival rate as the ratio of final and initial numbers of non-native species. Model scenarios include the four different trait profiles (high and low competitive and defensive strength), six levels of initial population size of non-natives, three levels of competition intensity, two levels of community density, two levels of mixture of the native community, and three levels of herbivore density (Table 3, Fig. 3). The competition index $\Theta$ describes the degree of intensity or asymmetry in competition and takes the values 0, 0.5 and 1 in our model. For the factor mixture of native community, native individuals are either mixed evenly or the percentage of native individuals of a species is derived from literature values about seedbank sizes (for more detail, see section Initialization). To reach our overall goal, we applied a binomial generalized linear model of the form:

*Survival of non-natives ~competitive strength of non-natives + defensive strength of non-native species + initial population size of non-natives + competition index $\Theta$ + native species density + mixture of native species + herbivore density +* all two-way interactions.

To account for overdispersion, we used a quasibinomial model. We simplified the generalized linear model by removing non-significant terms ($p \geq 0.05$) one by one, starting with the interactions, until only significant terms were left in the model. If a non-significant main term was part of at least one significant interaction term, we kept the main term in the model. This procedure led to the removal of the interactions between native species density and mixture of native species, defensive strength of non-natives and mixture of native species, and competitive and defensive strength of non-native species. We used a

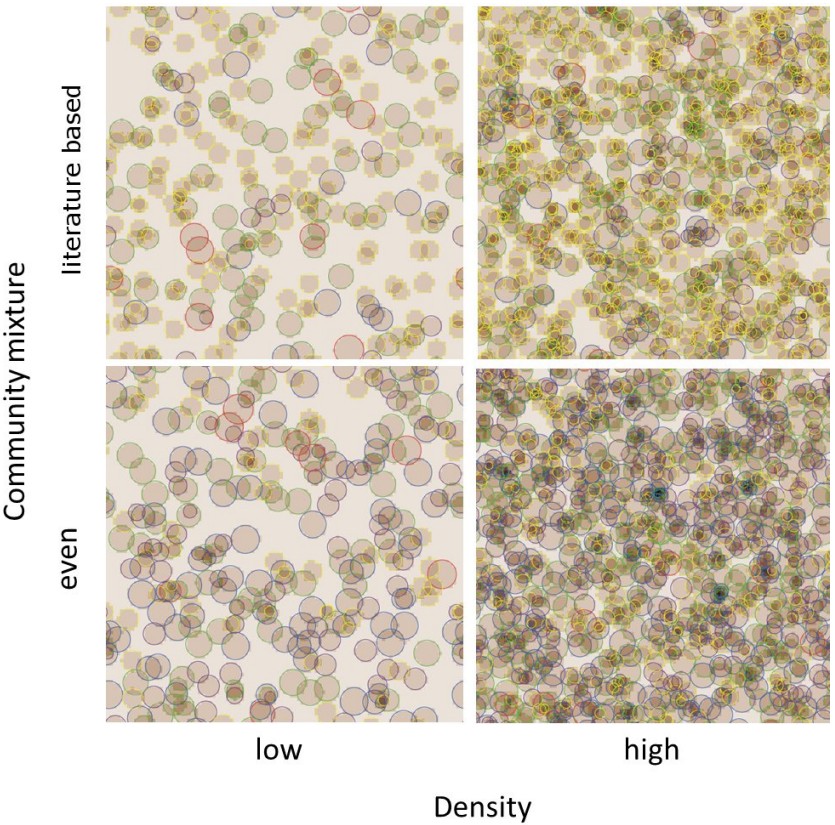

**Figure 3** **Screenshot of model communities after 14 days.** Model community is shown at varying initial community density (low: 308 plants, high: 1,108 plants) and varying mixtures of the native community (even: same initial density of all species; literature based: species initial densities differ based on reported seed bank sizes and germination rates). Species are color-coded: Red = non-native species, all other colors: native species, i.e., yellow = *A. capilaris*, green = *F. rubra*, purple = *H. perforatum*, blue = *P. lanceolata*.

chi-squared test to establish that the residual deviance was significantly reduced compared to the residual deviance of the null model. We used the software R version 3.3.3 (*R Development Core Team, 2017*) for data analysis.

## Sensitivity analysis

We conducted a sensitivity analysis to assess the relative importance of the input parameters with respect to model output. To this end, we used the same combinations of parameter values as in the Scenario analysis (Table 3) except for the parameters mixture of the native community and trait profile of the non-native species. Instead of these parameters, we ran sensitivity simulations with only one native species and one non-native species each, which were called species 1 and species 2. This approach made it possible to systematically assess the full range of possible trait values and not just the fixed trait profiles used for native and non-native species in the scenario analysis. The tested traits were seed weight, maximum biomass, repulsion of herbivores, seed mass per shoot mass, and germination probability for species 1 and species 2. To save computing time and still cover as much
trait parameter space as possible, we applied Latin Hypercube Sampling (*McKay, Beckman & Conover, 1979*) to assemble the trait profiles for the sensitivity analysis. We drew 15 samples from uniform distributions in the cases of repulsion (between 0 and 1) and germination probability (between 0.002 and 0.9) and from log2-distributions in the cases of seed weight (between 0.06 and 6), maximum biomass (between 3,000 and 30,000), and seed mass per shoot mass (between 0.02 and 6). The sampling distributions and ranges were determined based on our experience with the model simulations and always captured the range of the standard model parameterization. The values for each trait were randomly rearranged and then assembled into 15 trait profiles for species 1 and 15 trait profiles for species 2. Simulations were conducted with the full factorial combination of the remaining parameters from the scenario analysis (Table 3) for each of the 15 combinations of trait profiles of species 1 and 2. Model output was survival of species 2. The results of the simulations were analysed with a generalized linear model with quasibinomial errors in the same way as in the scenario analysis. This included model simplification and checking. To assess the sensitivity of the model output to the input parameters, standardized model coefficients were calculated by dividing model estimates by their standard errors. The absolute values of the standardized model coefficients were divided by their sum to obtain sensitivity values between 0 and 1. Sensitivities were sorted and plotted for comparability.

## RESULTS

All main and interaction terms in the generalized linear model contributed significantly to survival of non-native species ($p < 0.01$) except for mixture of native species (Table 4). The importance of competitive strength was very dominant and was a much stronger source of variation than defensive ability for survival of non-natives (main effects Comp and Def in Table 4, Fig. 4). Overall, weak competitors had a much lower survival rate than strong competitors. Although the populations of weak non-native competitors had an increased chance of persistence with an increasing initial population size they were not able to catch up with survival rates of strong competitors (interaction Comp:initNN in Table 4, Fig. 5). This was the case even though the survival of strong competitors was significantly negatively influenced by the population size of non-natives (interaction Comp:initNN in Table 4, Fig. 5).

Intensity of competition $\Theta$ had a marked negative effect on survival rates of non-native plants, especially on weak competitors (main effect Theta and interaction Comp:Theta in Table 4, Fig. 6). For weak competitors, average survival dropped by 80% when comparing equal share ($\Theta = 0$) and perfect size-symmetry ($\Theta = 1$). The decrease in survival rate was more severe in the even community mixture, as well as under high community density (interactions Comp:Mix and Comp:DensComm in Table 4, Fig. 6). Strong competitors were neither visibly affected by community mixture nor by increased population densities (Fig. 6).

In our model runs, we found that herbivores were only able to kill a plant in very early stages when they were very small. Herbivore damage increased survival of weaker competitors by a factor of up to 1.5 (main effect DensHerb in Table 4, Fig. 7). Weak

**Table 4 Effects of traits and community configuration on survival of non-native species.** Effects of trait values (competitive and defensive ability) and community configuration (herbivore density, community density and mixture, intensity of competition and initial number of non-natives) on survival of non-native species in a generalized linear model with quasibinomial errors presented as estimates of the effects and their corresponding standard errors, $t$-values and $p$-values. The Intercept corresponds to low competitive ability, low defensive ability, high community density and even community mixture. Asterisks indicate $p$-values smaller than 0.01 (**) or 0.001 (***).

| | Estimate | Std. Error | $t$-value | $p$-value | |
|---|---|---|---|---|---|
| Intercept | −2.43 | 0.020 | −118.68 | <0.001 | *** |
| High competitive ability (Comp) | 10.38 | 0.033 | 312.34 | <0.001 | *** |
| High defensive ability (Def) | −0.22 | 0.017 | −13.05 | <0.001 | *** |
| Initial number of non-natives (InitNN) | 0.006 | 0.000 | 75.87 | <0.001 | *** |
| Herbivore density (DensHerb) | 0.48 | 0.010 | 46.68 | <0.001 | *** |
| Competition intensity $\Theta$ (Theta) | −2.61 | 0.032 | −81.89 | <0.001 | *** |
| Low community density (DensComm) | 0.53 | 0.017 | 31.77 | <0.001 | *** |
| Uneven community mixture (Mix) | −0.005 | 0.016 | −0.34 | 0.73 | |
| Comp:InitNN | −0.014 | 0.000 | −141.82 | <0.001 | *** |
| Comp:DensHerb | −0.28 | 0.008 | −32.81 | <0.001 | *** |
| Comp:Theta | −2.67 | 0.021 | −127.50 | <0.001 | *** |
| Comp:DensComm | −1.20 | 0.014 | −82.55 | <0.001 | *** |
| Comp:Mix | −0.67 | 0.014 | −47.95 | <0.001 | *** |
| Def:InitNN | 0.0005 | 0.000 | 7.59 | <0.001 | *** |
| Def:DensHerb | 0.023 | 0.006 | 3.89 | <0.001 | *** |
| Def:Theta | 0.14 | 0.011 | 12.80 | <0.001 | *** |
| Def:DensComm | 0.064 | 0.010 | 6.63 | <0.001 | *** |
| InitNN:DensHerb | −0.0004 | 0.000 | −11.44 | <0.001 | *** |
| InitNN:Theta | −0.001 | 0.000 | −10.09 | <0.001 | *** |
| InitNN:DensComm | 0.0008 | 0.000 | 12.07 | <0.001 | *** |
| InitNN:Mix | 0.0005 | 0.000 | 7.29 | <0.001 | *** |
| DensHerb:Theta | −0.121 | 0.009 | −12.88 | <0.001 | *** |
| DensHerb:DensComm | −0.158 | 0.006 | −26.61 | <0.001 | *** |
| DensHerb:Mix | −0.016 | 0.006 | −2.71 | <0.01 | ** |
| Theta:DensComm | 0.544 | 0.016 | 33.49 | <0.001 | *** |
| Theta:Mix | 0.629 | 0.016 | 40.43 | <0.001 | *** |

competitors were also usually smaller than strong competitors. For strong competitors, this effect was the other way round, i.e., presence of herbivores was on average lowering the mean survival rate (interaction Comp:DensHerb in Table 4, Fig. 7). However, for strong competitors the decrease in survival rate was almost intractable, lowering survival by only 4% under complete size symmetry. Strongly defended species with 70% probability of repulsion only had a minor advantage in survival over weakly defended species with 40% probability of repulsion (main effect Def in Table 4).

The sensitivity analysis showed that the model output survival of species 2 was most sensitive to the interaction between competition index $\Theta$ and seed weight, then to maximum biomass, the interaction between the seed weights of the two competing species, the interaction between seed weight and germination probability of the same species, and
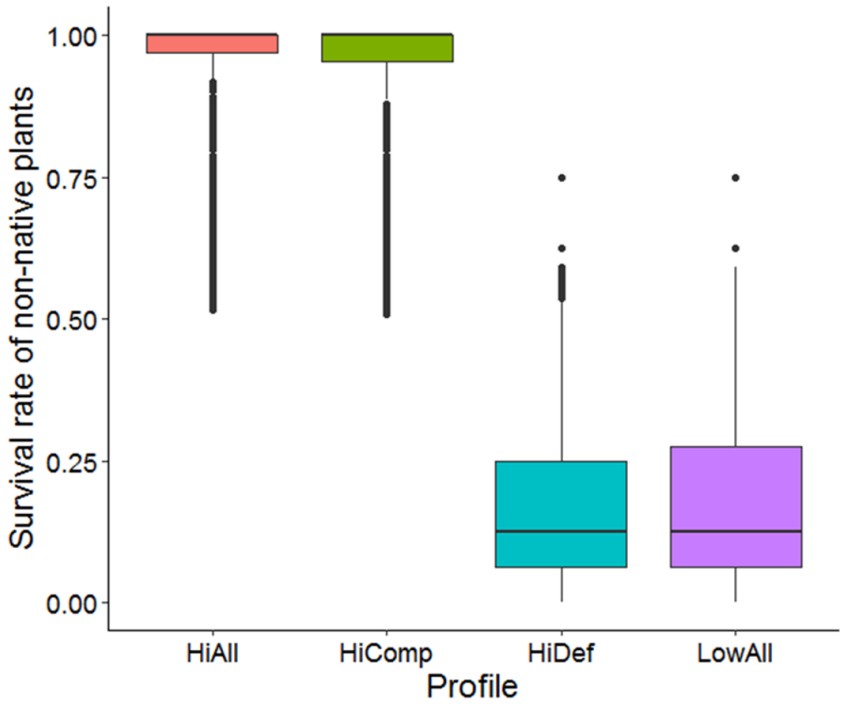

**Figure 4** **Survival rates of non-native plants with different trait profiles.** The trait profiles are: high competitive and high defensive (HiAll), high competitive and low defensive (HiComp), low competitive and high defensive (HiDef), and low competitive and low defensive (LowAll). Values were averaged over all scenarios.

the interaction of seed weight of species 1 with maximum biomass of species 2 (Fig. 8). Germination probability and seed weight were the next most important inputs for determining survival of species 2, together with the interaction between seed weight and repulsion of herbivores, competition index $\Theta$, and repulsion of herbivores. Overall, species traits and their interactions dominated the sensitivity ranking, whereas scenario parameters such as herbivore density, initial population size of non-natives and native species density were less important. Only the competition index $\Theta$ had a large, mostly indirect influence on survival of species 2 via interactions with species traits. In contrast to biomass- and competition-related parameters, herbivory-related parameters such as herbivore density and repulsion of herbivores played relatively minor roles for survival of species 2.

## DISCUSSION

With our model, we investigated the establishment success of different types of generic non-native plant species in a resident native community. We targeted three aspects that might influence establishment success: trait profile of the non-native species, biotic pressures of the resident community, and the interactions between traits and pressures. The sensitivity analysis showed that model outputs were much more sensitive to species traits and interactions between species traits and biotic pressures than to biotic pressures

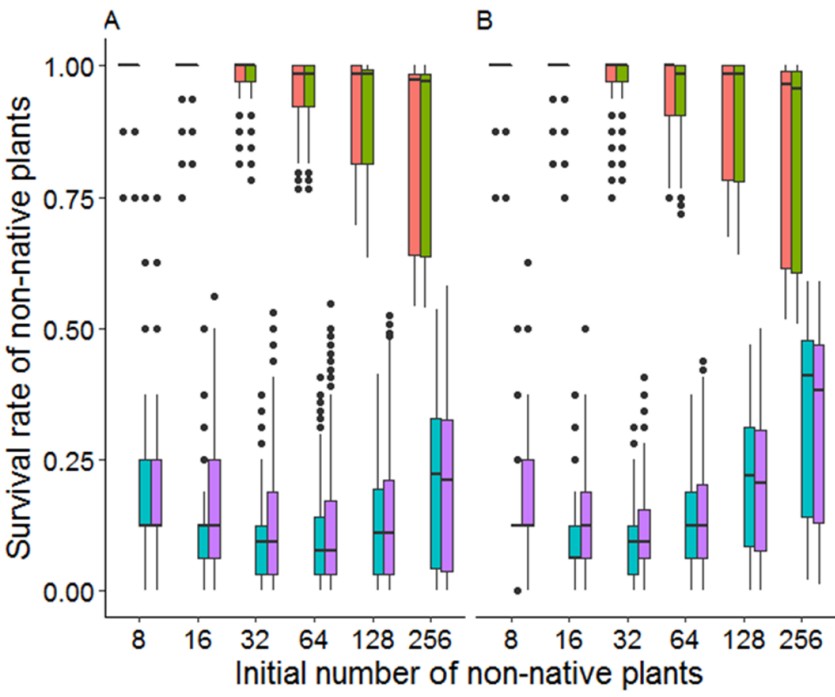

**Figure 5** **Survival rates of non-native plants under different invasion levels in high density (A) and low density (B) native communities.** Invasion level corresponds to initial population sizes of 8, 16, 32, 64, 128 and 256 individuals of the non-native species. Non-natives are split into the following trait profiles (in each block from left to right): high competitive and high defensive (HiAll, orange), high competitive and low defensive (HiComp, green), low competitive and high defensive (HiDef, blue), and low competitive and low defensive (LowAll, purple).

alone. Thus, care should be taken in the choice of experimental and model species and the traits they represent.

Competitive pressures and competitive traits exerted a much stronger influence on establishment success than pressures and traits related to herbivory. This was supported by scenario analysis and sensitivity analysis. Strong competitive traits were negatively correlated with initial density of non-natives in their effect on survival rates while a low initial number of non-natives with strong competitive ability resulted in high survival. We expect this might be due to a potential release effect from competition. These scenarios may for instance reflect distant satellite populations or the very edge of the expansion front, because there is evidence that population density on the range borders can be lower than in the core range (*Brown, 1984*; *Maggini et al., 2011*). High survival in these conditions may translate to an effective range shift, and this is more likely if long distance dispersal is included. However, a higher initial number led to a decrease of survival rate for strong competitors. An explanation may be that the high density of individuals with strong competitive traits led to a milieu of competitive stress, provoking intra-specific self-thinning effects (*Morris, 2003*). In our model setting, there was no evidence of complete extinction of non-native species due to competitive stress. Thus, the high intra-specific competition decreased individual survival, but did not decrease establishment success of
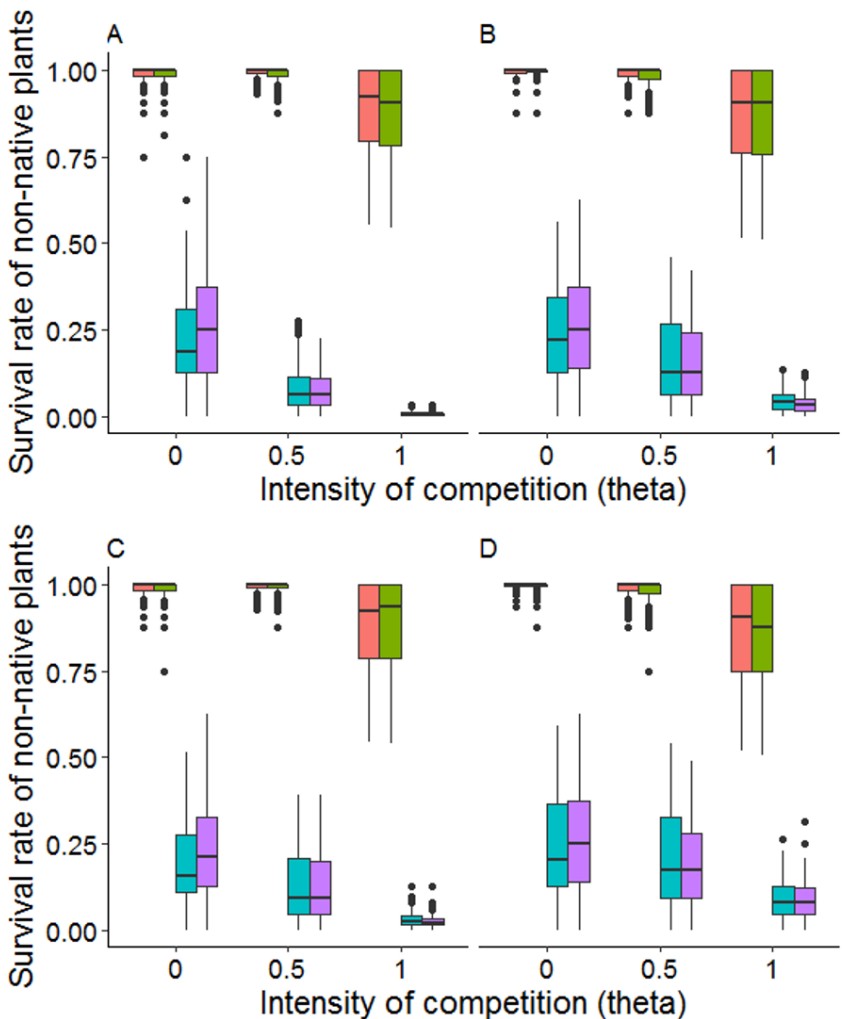

**Figure 6 Effect of the intensity of competition Θ on survival rates of non-native plants in different configurations of the resident community.** (A, C) show high density of natives, (B, D) show low density of natives. Note that different initial numbers of non-natives are not separated in this figure. (A, B) show an even mixture of natives in the initial community, (C, D) show a mixture based on literature values of seed bank sizes. With $\Theta = 0$, resources are shared among competitors regardless of their biomass, with $\Theta = 1$, resources are shared proportionally to the biomass of the individual competitors. $\Theta = 0.5$ reflects an intermediate stage. Non-natives are split into the following trait profiles (in each block from left to right): high competitive and high defensive (HiAll, orange), high competitive and low defensive (HiComp, green), low competitive and high defensive (HiDef, blue), and low competitive and low defensive (LowAll, purple).

the populations. It would thus probably not strongly impede range expansion of non-native species. Rather, this may contribute to stabilizing the range expansion.

Populations of weak competitive plants showed much lower survival at lower densities than strong competitors, implying a much lower chance of long-term establishment. Survival of weak competitors required very high initial numbers to exceed survival at low initial numbers of individuals in the population. This might be due to the sheer mass of

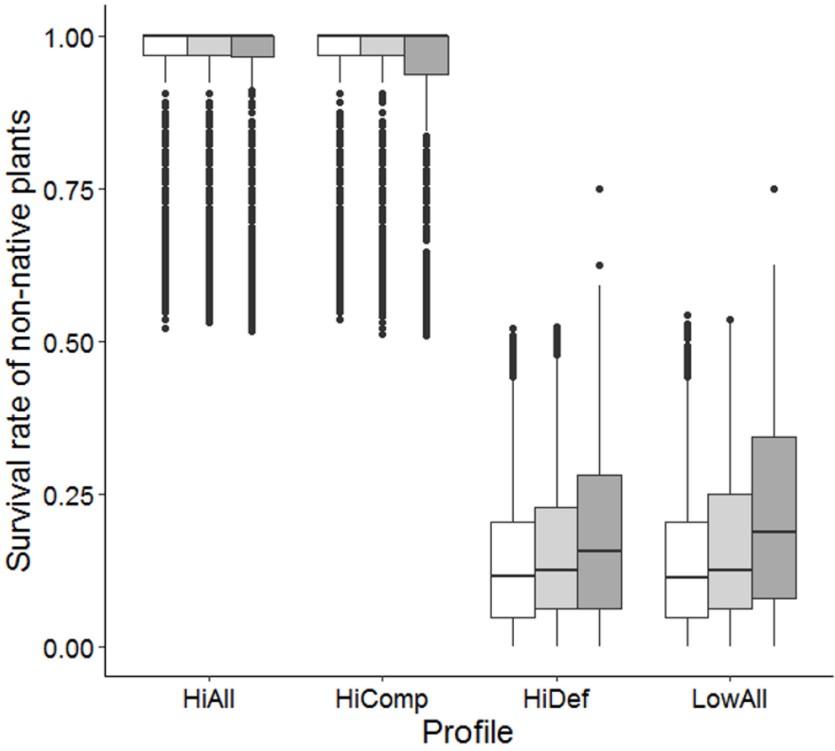

**Figure 7 Survival rate of non-native species at different herbivore densities.** Herbivore densities: no herbivores as control scenario (white bars), 3.2 herbivores per m$^2$ (light grey bars), and 18.75 herbivores per m$^2$ (dark grey bars). Non-natives are split into the following trait profiles: high competitive and high defensive (HiAll), high competitive and low defensive (HiComp), low competitive and high defensive (HiDef), and low competitive and low defensive (LowAll).

non-native individuals that competed with the native community. In the most extreme setting 256 non-native individuals faced 300 native competitors. Thus, even if native species were the stronger competitors, a high number of weak non-native competitors might be able to overcome the biotic resistance, a trend that has already been observed in studies of intercontinental invasions (*Lockwood, Cassey & Blackburn, 2005*; *Simberloff, 2009*). This effect was clearly observable although there was only a small difference between seed weight of weak non-native species and native competitors. We expect the effect to be even more pronounced if the difference in seed weight was larger.

The advantage of strong competitors compared to weak competitors persisted in community settings with more intense community-borne competitive stress, i.e., at higher community density and species mixtures with a higher proportion of strong competitors. Higher community density results obviously in a higher number of competitors for each individual plant and thus in most cases a lower amount of resources that can be captured by any plant. For the community mixture, an even number of individuals of all species led to a community with a much larger proportion of strong competitors, i.e., *P. lanceolata*, the largest native species in our setting, than in the seed bank-based mixture. However, since strong non-native competitors were characterized by twice the seed weight of *P. lanceolata*
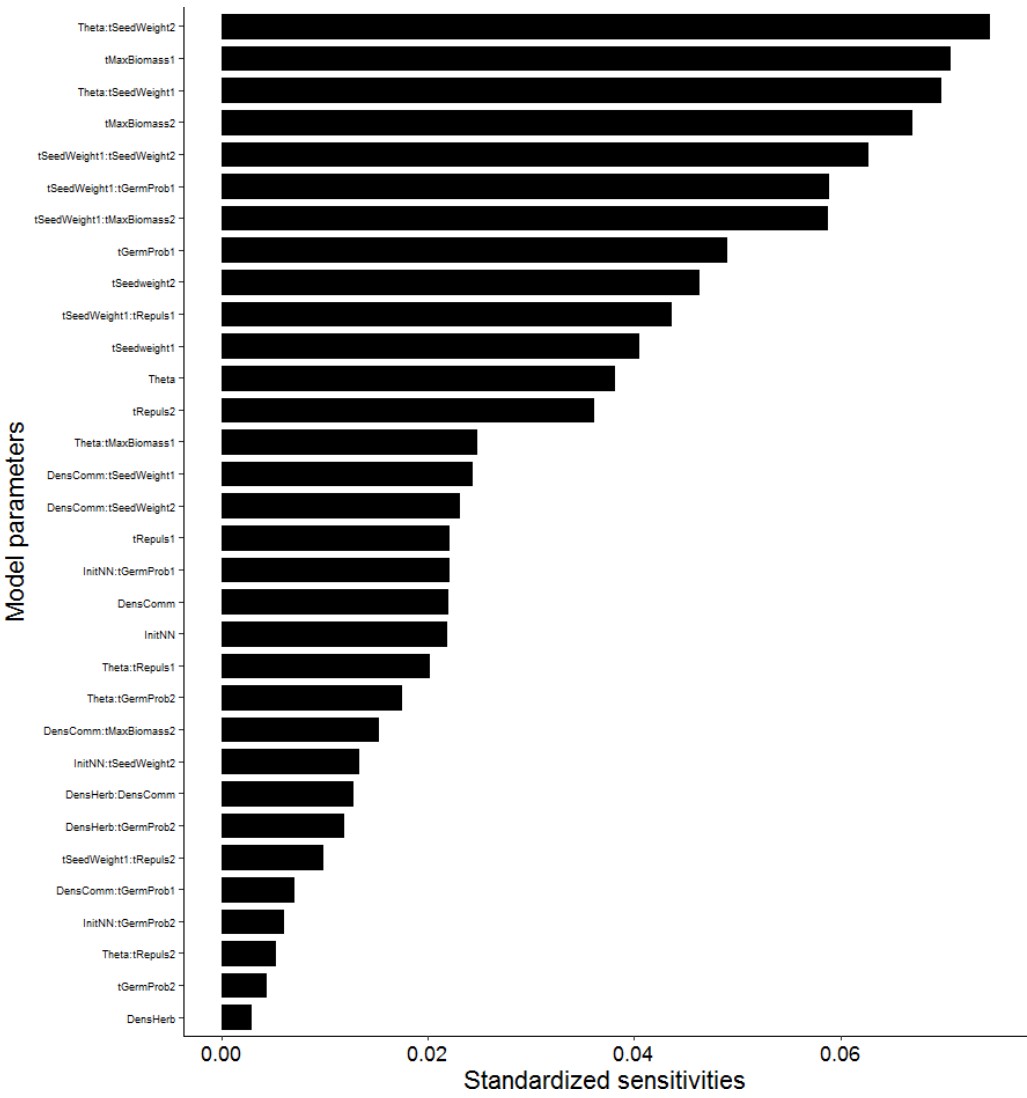

**Figure 8** **Standardized sensitivities of model output to model parameters.** Standardized sensitivities were a result of a generalized linear model of simulation results after statistical model simplification. Simulations were conducted with two species each, which corresponded to "native" and "non-native" species in the scenario analysis, but whose trait profiles were assembled here independent of real species (trait names ending with "1" or "2"). Model output was survival of species 2. Model parameters covered biotic pressures, traits (names starting with "t"), and their interactions. Biotic pressures included competition index (Theta), community density (DensComm), initial number of "non-natives" (InitNN), and density of herbivores (DensHerb). Traits included seed weight (tSeedWeight), maximum biomass (tMaxBiomass), germination probability (tGermProb), and repulsion of herbivores (tRepuls).

they were probably not massively impacted by numbers of competitors or community mixture. We observed this effect in the preceding empirical experiment as well, where strong non-native competitors were massively dominating the native community (*Radny et al., in press*).

These results indicate that a range shift should be more effective for such plants that are strong competitors relative to species of the receiving community, provided that their seeds can reach a novel habitat. For intercontinental invasions competitive strength is one of the major explanations of invasive success (*Vilà & Weiner, 2004*). In the context of climate-change induced range expansion, this might become just as important or even more important, because changes in the microclimatic regime of habitats beyond current range borders may weaken the currently strong resident competitors and thus increase invasibility of communities (*Alexander et al., 2016*; *Bauer, 2012*; *Stanton-Geddes, Tiffin & Shaw, 2012*).

However, the very low survival rates of weak competitors in our model may overestimate the negative impact of community competition on weak competitors in reality. For instance, in our preceding experiment (*Radny et al., in press*), drop-out rate of weak competitors was almost zero while in the model weak competitors responded very drastically to increased competitive pressure from the community in form of increased density, mixture and intensity of competitive asymmetry ($\Theta$). Probably, in our model, we underestimated the abilities of weak competitors to avoid or tolerate competitive pressure from other species. This might be partly due to the implementation of competition with the Zone-of-influence approach. For theoretical models of competition between plant individuals, the Zone-of-Influence approach has been used many times at different degrees of complexity (e.g., *Berger et al., 2008*; *Lin et al., 2012*; *Weiner & Damgaard, 2006*). Despite several simplifications, it is a straightforward and comparatively easy method to investigate competition (*Berger et al., 2008*). However, most of these models address monocultures of species and thus implement the same type of interaction, i.e., degree of asymmetry $\Theta$, for all individuals. Interspecific interactions may be different though from intraspecific interactions due to many different mechanisms. This may not only imply differences in interspecific and intraspecific $\Theta$, but also different $\Theta$-values depending on the identity of the focal species (*Connolly & Wayne, 1996*). Such mechanisms include for example allelopathy (*Bais et al., 2003*) or adaption to the competitive disadvantage, e.g., development of shade tolerance in trees (*Dislich, Johst & Huth, 2010*). Additionally, following the parsimony principle we have not yet considered facilitative interactions in this model, although there are potentially strong impacts of facilitative interactions in plant communities (e.g. *Lin et al., 2012*). Of course, parameterizing different competition types for all possible interaction partners in our five-species system would require a lot of data, which were not available in our case, and has also been attempted in only very few comparable cases thus far.

Thus, we strongly advocate for the extension of multispecies models to incorporate different forms of neighbourhood interactions not only as negative (competition) or positive (facilitation) interactions, but also accounting for different intensities of inter- and intraspecific interactions. This approach will require enhanced efforts in the collection of adequate data for parametrization, but we expect a much better understanding of multispecies systems from such approaches (*Svenning et al, 2014*).

The effect of herbivory was comparatively small in our model. Accordingly, defensive strength did not play an important role for survival. This may be due to the indirect influence of biomass on survival via its effect on plant growth (Fig. S1) combined with the

fact that model herbivores only consumed absolute amounts of biomass. This means that large plants suffered relatively less from herbivory than small plants. In extra simulations, where herbivores consumed relative amounts of plant biomass, defence traits were much more important for survival than in the standard simulations (Supplemental Information 1). Herbivory has been reported to influence range expansion and invasions, i.e., in spatial pattern and speed (*Fagan et al., 2005*; *Herrero et al., 2016*), yet it is unlikely that herbivory may entirely block establishment of novel species (*Jeschke et al., 2012*; *Levine, Adler & Yelenik, 2004*). However, although herbivory as a single factor may not pose a hard barrier to establishment, studies found herbivory to be an important interacting effect under competition through altering the competitive impact of individuals (*Huang et al., 2012*; *Kim, Underwood & Inouye, 2013*; *Kuijper, Nijhoff & Bakker, 2004*). Our model results support such an interaction between herbivory and competition, where weak competitors showed increased survival under herbivory, especially with higher densities of herbivores. We suspect that weak competitors benefited from being small relative to their neighbours—either due to initial small size or due to cumulated negative effects on biomass gain from competition. As sharing of the contested resource can depend on the relative biomass of the competitors, loss of biomass due to herbivory can reduce the resource capture of strong competitors, so that more resources are left to neighbouring weak competitors than in scenarios without herbivory. In our model, the minimum amount of resource uptake for maintenance of metabolism and thus survival is directly related to current biomass of the individual (*Brown et al., 2004*). Thus, even a relatively small increase in resource capture can increase survival of smaller plants.

Of course, our model did not capture the full complexity of possible herbivore impacts on plant distribution and range expansion. Body size of herbivores and timing of herbivory have been shown to differentially affect plant biodiversity (*Kim, Underwood & Inouye, 2013*; *Olff & Ritchie, 1998*), as well as presence or absence of specialist herbivores (*Joshi & Vrieling, 2005*; *Lakeman-Fraser & Ewers, 2013*). Our extra simulations on relative herbivory (Supplemental Information 1) indicate that it may be worthwhile to explore a greater range and resolution of herbivory implementations once the respective data become available for parameterization. Future model extensions could reflect these factors as well as plant internal mechanisms such as compensatory growth (*Lu & Ding, 2012*) and increase of defence mechanisms (*Strauss & Agrawal, 1999*). However, even with our simple model design, we found a significant effect of herbivory—although not the expected global decrease of survival, but an indirect effect through harming the competitors. This indicates that herbivory effects may sometimes be overseen when they are not turning out as expected and that this might also be a reason for contradicting results of similar studies (*Jeschke et al., 2012*; *Levine, Adler & Yelenik, 2004*). For further developments of local competition models in a community context, we advocate to develop approaches that include tolerance strategies. In the frequently used ZOI approach this could be realized by an asymmetry index that is sensitive to the identity of interaction partners.

## CONCLUSIONS

We conclude that traits related to competitive strength of species can change the effectiveness of biotic resistance from resident competitors and should be taken into account when attempting to predict establishment success of range expanding species. Where the impact of herbivores is of minor importance, strong defence traits do not result in an apparent advantage as compared to weak defence. However, herbivory might have a stabilizing effect on competition and thus should not be neglected when analysing range expansion dynamics. This model may serve as a basis for future large-scale models, where dispersal should be considered as a third important trait to describe range expansion.

### Funding

This work was supported by the Deutsche Forschungsgemeinschaft DFG (ME_3575/2-1). The funders had no role in study design, data collection and analysis, decision to publish, or preparation of the manuscript.

### Grant Disclosures

The following grant information was disclosed by the authors:
Deutsche Forschungsgemeinschaft DFG: ME_3575/2-1.

### Competing Interests

The authors declare there are no competing interests.

### Author Contributions

- Janina Radny conceived and designed the experiments, performed the experiments, analyzed the data, prepared figures and/or tables, authored or reviewed drafts of the paper, approved the final draft, developed model concept and code.
- Katrin M. Meyer conceived and designed the experiments, analyzed the data, contributed reagents/materials/analysis tools, prepared figures and/or tables, authored or reviewed drafts of the paper, approved the final draft, developed model and concept and reviewed model code.

### Data Availability

    The raw data and model code are provided in Supplemental Files.

### Supplemental Information

Supplemental information for this article can be found online at http://dx.doi.org/10.7717/peerj.5342#supplemental-information.

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
