# Peer review of "The role of biotic factors during plant establishment in novel communities assessed with an agent-based simulation model"

_PeerJ, doi:10.7717/peerj.5342_

## Round 0.1 · original submission · Major Revisions

Please pay close attention to the recommendations for improving the readability and understandability given by the reviewers. Performing the sensitivity analysis will also be important so that readers can evaluate the usefulness of your analysis.

·

Basic reporting

no comment

Experimental design

no comment

Validity of the findings

no comment

Additional comments

Dear Authors,
With pleasure I have read your interesting and very well-written manuscript. I like your work, your embedding in the literature and the conclusions you draw from your computational experiments. I have some questions and comments, and a, what I think, important suggestion that I’d like to hear your thoughts on.
To start with the latter:
I think the strong difference between survival rates of competitors and defenders could be due to an ‘imbalance’ in the model: plant growth (equation 2) depends on biomass, both through the zone of influence and the ratio with maximum biomass. Biomass does have a contrasting effect on plant growth through both parts of the equation. It would be nice to see the graphs for some different values of biomass for overall plant growth in the appendix, to get more feeling for what the equation implies. In addition, herbivores damage plants by eating a certain amount of biomass, equal for big and small plants, and don’t spend more time on one than the other. This means that big plants suffer relatively less from the same number of herbivores. In all, biomass is what drives the plant’s survival probability.
(btw: I seem to miss the values for rgr in the paper, are they different for different species?)
And related to this a question: during a run, no new plants appear in the plot, right? So survival rates are a maximum of 1? This means I guess that if a good competitor outcompetes its neighbour (which would usually be replaced by a new competitor), it’s after that care-free, with a bigger zone of influence than before.
If all this means that large biomass is relatively important in the model, it also makes sense that defence values don’t matter: if herbivores are not important then defending against them doesn’t either. We do see trade-offs in the native plant traits, but under model simplification one benefit can overwhelm the other.
(Speaking of which: it seems that 3 native plants have lower seed weight than what is called ‘weak’ for non-natives?)
Other comments:
I think the study (apart from some minor issues) is very well-written and explained. Sometimes even a bit apologetic, as if using a model needs special conditions or warrantees. In line with this:
-ln. 64-67: turn around: non-successful expansions cannot be tracked in the field, and we need full control to test certain hypotheses here, so we use a model.
- ln. 68: a greenhouse is not reality either, you do not need to apologise. rephrase please.
- 74: this should be the view of any person interested in range expansions, leave out the modeller’s view.
- 101: you can make this statement in the discussion, no need to apologise here.
- 111: first mention of dispersal. I value that you show the bigger picture and maybe some reviewer asked for including all this. But I find it confusing to talk about dispersal when it is actually not included in the model..
-130 vs 138: 3-dimensional trait vector vs 4 plant traits?
-139: plants always hatch from seeds :)
-176: so 1 cm2?
-185: what is the purpose of stochastic density of herbivores: is this also not a reason that biomass is valued over other traits? If a herbivore remains in a spot, it would do so on a large plant I guess..?
-220: I guess the same comment as for dispersal: I value that you explain how you actually had herbivores move during a simulation, but for this purpose I would just leave out how you solved this technically and just say they chose a new random position each day.
-233: another one: if you remove herbivores that cannot eat on defended plants, without them bothering other undefended plants with it, this gives a relative advantage for undefended plants. In real life I guess the herbivore would continue searching for a plant that it can eat.
- I don’t understand how you get from eq 3 to eq 4. Maybe something for the appendix (‘appendix for the stupid’ :)
- 271: but theta was between 0 and infinity? (ln 261)
- 289: note that = so
- 292-294: I don’t understand why you do this: is this a trade-off: strong competitors likely maintain more biomass but set less seed per amount biomass?
-304: parameterization?
-334: omit ‘from reality’
-344: so no new plants during the simulation? Which would make survival rates >1
-365: to catch up
-388: because strong competitors end up competing with each other?

·

Basic reporting

The article is generally well-written and follows the ODD protocol structure. This is slightly different to a traditional research article layout but it is commonly used for individual based models and is appropriate here. The authors provide a good background for the study in the introduction and the references cited are appropriate.

However, on occasion the language used is a little hard to understand. Some sentences are quite long and convoluted, and other times the grammar is not quite right, common mistakes include e.g.“are interacting with” should be “interact with” (line 175), “is representing” should be “represents” (line 146). I provide some further examples at the end of this section. In general, I would recommend that a native English speaker provide feedback on the draft before publication.

I also think it would be good to make clear early on that this model has been parameterized using data from a greenhouse experiment, along with some basic information about the experiment – even though the authors provide a copy of this manuscript (thank you!), just some general details e.g. when the experiments took place, the basic design, what species, etc. would be useful to know early on rather than mid way through the paper. At the moment this information is drip-fed throughout the paper, which makes it a little hard for the reader to keep track of.

The raw data has been shared (the netlogo model), however I think that the article could be improved by sharing the R code and model outputs as described in lines 352-358. I talk a bit more about this in section 3 of the review.

Some examples of where language & grammar could be improved:
Line 27-28 and 30-31 - sentences are quite hard to follow, can you replace “as compared to” with “than” perhaps?
Line 43 – it is not clear who “they” refers to here
Line 46 – you use both the terms “range shift” and “border shift” – I suggest for consistency only using the first of these terms as multiple terms can become confusing
Line 48 – suggest “include, for example, ..”
Line 130-131 – Can a vector be 3D? Should this be matrix instead?
Line 162 – Wording is a bit confusing. Recommend “resident herbivores and competitors” instead
Line 128,188 – “Repulsion” instead of “repulse”?
Line 240-241 – this seems like repetition of earlier “interactions” section – probably doesn’t need to be cited twice and don’t need to re-define acronym

Experimental design

The research fits well within the aims and scope of PeerJ, and the research question is well defined and is of great relevance in invasion biology. The introduction is well laid out and explains the state of knowledge well, as well as placing the study in a wider context. However, I think it would benefit from explicit hypotheses or aims at the end, as the current sentence at the end is a little vague and does not strongly describe what knowledge gaps the research is filling.

The methods are extremely detailed and I commend the authors on their use of the ODD protocol and for providing their model. I can confirm that the model works and that the code is well-commented and laid out so that a third party can understand it easily. The technical quality of the work is high and I cannot find fault with it.

My only suggested improvement here is that the binomial GLM models could be more fully described in the text (i.e. replace “and all two-way interactions” with the variable names), as well as how they were assessed (e.g. were data normalized/standardized, were residuals checked, how was the model assessed for degree of fit etc). This, along with providing the R code for these models, would greatly help improve reproducibility.

Another minor comment is that the info tab of the netlogo model is blank when I open it on my computer – I’m not sure if this is because I’m using a newer version of netlogo or because the info tab hasn’t been filled out. If the latter, I recommend doing so to make the model more user-friendly.

Validity of the findings

The data appear to be robust, and the conclusions are well linked to the introduction and research question. However, it is difficult to fully assess the robustness of the results as there is no sensitivity analysis of the model. I would recommend running a full sensitivity analysis for all parameters and presenting these results in a table, as it is difficult to assess the validity of the model without this information.

I think that before publication it would be good to expand the results section – personally I would like to see the results of a sensitivity analysis, the results of the scenarios, as well as the results of the binomial GLMs described in the text. Currently, the results text is a bit vague and lacking in detail, and as such it is hard to assess the robustness of the conclusions even though the authors provide some of this information in figures and tables.

Additional comments

I very much enjoyed reading and reviewing this paper and have no doubt that it will be up to the standards of PeerJ after some minor editing as described in the previous sections. Thanks to the authors for providing clearly-commented code and for following the ODD protocol, this is something I really think we need more of in the modelling literature!

My main suggestions for improvement (in order of importance) are:
1. Provide sensitivity analysis results for model parameters
2. Expand upon results section text and description of GLMs in methods
3. Improve grammar and reduce wordiness throughout

My recommendation (minor revisions) is based on the assumption that the authors have already done a sensitivity analysis during model construction. If this is not the case it could be quite a significant undertaking, if that is the case and if the authors decide not to include a sensitivity analysis, I think this needs to be addressed and justified in the methods as it is fairly standard practice.

My other comments are just minor suggestions. Thanks again to the authors for their well-laid out paper, I had a lot of fun playing with your model, it's very nicely done!

---

## Round 0.2 · accepted · Accept

Thank you for making clear and comprehensive responses to the reviewer's comments, The major comments that you were asked to address were to provide better explanation of what you did (I confirm hat was changed) and also to perform a sensitivity analysis on some parts of the model (which I also confirm was completed). Your replies to each of the comments are very comprehensive and I felt that you addressed all the prior comments correctly.

#